# Unveiling GABA and Serotonin Interactions During Neurodevelopment to Re-Open Adult Critical Periods for Neuropsychiatric Disorders

**DOI:** 10.3390/ijms26125508

**Published:** 2025-06-09

**Authors:** Emanuela Beretta, Gianmarco Cuboni, Gabriele Deidda

**Affiliations:** 1Department of Biomedical Sciences, University of Padua, via Ugo Bassi n. 58/B, 35121 Padua, Italy; 2Faculty of Medicine & Dentistry, Queen Mary University of London, Malta Campus, VCT 2520 Victoria, Malta; 3Padua Neuroscience Network, University of Padua, 35121 Padua, Italy

**Keywords:** neurodevelopment, plasticity, GABA, serotonin, cognition, neurodevelopment, adult plasticity, neuropsychiatric disorders

## Abstract

The mature brain is the result of a complex neurodevelopmental process resulting from interweaved mechanisms and involving early genetic and microenvironmental factors shaped by patterns of spontaneous electrical activity. During postnatal development, the immature brain undergoes experience-dependent structural and functional shaping and modifications during critical period (CP) time windows to achieve the full maturation of brain functions. Plasticity is higher during neurodevelopmental CP windows and is limited in the adult brain, including during neuropsychiatric disorders. Notably, the neurotransmitters γ-aminobutyric acid (GABA) and serotonin are two fundamental players controlling and modulating, respectively, brain plasticity in the developing and adult brain. Therefore, acquiring insights into the roles played by GABA and serotonin in regulating CP plasticity might hold potential for pharmacologically re-opening CP windows in adult life, with the aim of providing therapeutic intervention for neurological and neuropsychiatric disorders.

## 1. Introduction

During prenatal and postnatal development, countless calibrated events occur in the brain to achieve proper neuronal migration and positioning and neuronal network wiring [1,2]. As neurodevelopment begins, neuronal progenitor cells—already committed to a neuronal fate—migrate to their final location within the brain, establish synaptic connections, and accomplish neuronal differentiation. During postnatal life, experience-dependent neuronal activity is critical for synapse reduction and refinement, eliciting plastic events that shape the neuronal networks structurally and functionally [3,4,5]. Of note, pre- and postnatal neurodevelopment is tightly regulated by the neurotransmitter γ-aminobutyric acid (GABA), and by 5-hydroxytryptamine (5-HT), better known as serotonin [6,7].

GABA is one of the most abundant neurotransmitters in the mammalian brain. It acts by binding ionotropic (GABA_A_) and metabotropic (GABA_B_) receptors. GABA_A_ receptors are ion channels permeable to chloride anions (Cl^−^), accounting for phasic (synaptic) and tonic (extra-synaptic) currents [8,9]. The net flow direction of Cl^−^ in neurons through GABA_A_ receptors depends on the expression level and activity of two main chloride cotransporters: (i) the Na^+^/K^+^/Cl^−^ cotransporter (NKCC1) and (ii) the K^+^/Cl^−^ cotransporter (KCC2). Dysfunctions in NKCC1 and KCC2 are described in a number of neuropathological conditions, including neurodevelopmental disorders—for example, Down syndrome, autism, and schizophrenia spectrum disorders—and neurological disorders—for example, spinal cord injury and Alzheimer’s disease [10,11,12].

The action of GABA during neurodevelopment is widely described, but serotonin also plays a key role. Serotonin binds to different subtypes of receptors; among these, six (5-HT_1-7_) are G protein-coupled receptors and only one (5-HT_3_) is a ligand-gated ion channel [4,13,14]. Interestingly, during embryonic life, serotonin acts as a morphogenic factor [15,16,17] while, later, once development is complete, it fine-tunes countless human behaviors [14,18,19]. Consequently, dysfunctions in the serotonergic system—in other words, in GABA—have also been shown to be involved in a number of neurodevelopmental and neurological disorders, including schizophrenia, autism spectrum disorders, and depression [13,14]. For these reasons, serotonin is commonly considered a main therapeutic target [20,21,22].

Importantly, GABA and serotonin not only act independently, but also act in concert to control and modulate plasticity during prenatal [23,24] and postnatal neurodevelopment [4,7,25]. Once the critical period (CP) plasticity windows for brain maturation close, brain plasticity wanes significantly, limiting the possibility of therapeutic interventions in adult age [26,27,28,29].

In this review, we will first describe the roles of GABA and serotonin throughout neurodevelopment and adult life. Then, we will highlight recent findings in the literature on the effects of psychedelic drugs on re-opening therapeutic plasticity windows in patients with neuropsychiatric disorders.

### 1.1. The GABAergic System

GABA is the main hyperpolarizing and inhibitory neurotransmitter in the adult brain [30]. However, it has a depolarizing and excitatory action during neurodevelopment and in a number of physiological processes (for example, parturition, circadian clock regulation, hibernation, etc.), as well as in many brain pathologies [7,30,31].

During GABAergic neurotransmission, the enzyme glutamic acid decarboxylase (GAD) synthesizes GABA from glutamic acid [32]. GABA is packed into presynaptic vesicles by vesicular GABA transporters (VGATs) using electrical and chemical gradients [33] (Figure 1). Upon the depolarization of the presynaptic terminals by action potentials, GABA is released into the synaptic cleft, where it binds to the GABA_A_ and GABA_B_ receptors [7]. The GABA_A_ receptors are heteropentameric ionic channels formed through the assembly of five subunits belonging to 19 different classes (α_1−6_, β_1−3_, γ_1−3_, δ, ε, θ, π, and ρ_1−3_) (Figure 1).

As mentioned, GABA_A_ receptors are Cl^−^-permeable ion channels [34,35]. Cl^−^ flows through the plasma membrane, and its effect on neuronal potential [36] depends on the intracellular Cl^−^ concentration. As mentioned earlier, two main chloride cotransporters control intracellular Cl^−^ concentration and, hence, set the equilibrium potential (E_Cl_, also known as Nerst potential) for Cl^−^ anions: (i) the Na^+^/K^+^/Cl^−^ cotransporter NKCC1 [37], which pumps Cl^−^ into the neuron, and (ii) the K^+^/Cl^−^ cotransporter KCC2, which pumps Cl^−^ outside the neuron. Notably, NKCC1 is expressed more during early development—accounting for depolarizing and excitatory GABA action—and KCC2 is expressed more during adult life—accounting for hyperpolarizing and inhibitory GABA action [6,38,39].

GABA_B_ receptors are G protein-coupled metabotropic receptors located both in the central and the peripheral nervous system [40]. Upon the binding of GABA to GABA_B_ receptors, a conformational change enables interaction with the heterotrimeric G protein that dissociates into Gα and Gβγ subunits. Notably, on the postsynaptic side, GABA_B_ receptors activate inward-rectifying K^+^ channels (GIRK), leading to hyperpolarization and a slow inhibitory current that dampens dendritic calcium spike backpropagation and shunts ongoing currents [40,41].

### 1.2. The Serotonergic System

Serotonin is a biogenic monoamine functioning both as a hormone [42] and as a neurotransmitter, with numerous roles in the central and peripheral nervous systems [43].

Different studies have revealed its major role in fine-tuning the whole spectrum of human behaviors, including emotions, anxiety, and depression [19,24,43,44,45]. Pharmacological studies have confirmed serotonin as an important mediator for motivation, cognition, and reward, making it as significant as the neurotransmitter dopamine [46], and positioning its receptors as a key target for drug discovery [47,48].

The anatomical distribution of serotonergic neurons was described for the first time in 1964, when Dahlström and Fuxe (1964) identified nine clusters of serotonin neurons—alphanumerically indicated as B1–B9—within the brainstem [49,50]. These neurons already cluster together during early embryonic development into a caudal (B1–B3) and a rostral division (B4–B9) [13,17,50] (Figure 2a). The rostral group (midbrain *raphe*)—enclosed within the mesencephalon and rostral pons—primarily projects to the forebrain, whereas the caudal group (medullary *raphe*), situated in the *medulla oblongata*, projects to the *caudal medulla* and spinal cord [19,51]. In particular, the rostral group includes the *nuclei pontine central oralis* (B4), the *median raphe* (B5), the *dorsal raphe* (B6–B7), *caudal linear nucleus* (B8), and *medial lemniscus* (B9), that collectively project to the forebrain and the brainstem [52,53]. The caudal group, instead, includes the nuclei *raphe pallidus* (B1), the *raphe obscurus* (B2), the *raphe magnus* (B3), the *area postrema*, and the *lateral medulla*, all giving rise to descending serotonin projections directed toward the cerebellum and the spinal cord [50,54] (Figure 2b).

Serotonin synthesis occurs in *raphe neurons* in the brainstem and also in the enterochromaffin cells in the gut—where most of body’s serotonin is synthetized [55]. Given that serotonin cannot cross the blood–brain barrier (BBB) [43], the two biosynthetic processes occur independently from each other. In addition, two homologous tryptophan hydroxylase (TPH) enzymes exist: one selectively expressed in the serotonergic *raphe* (i.e., TPH 2), and the second localized in myenteric gut neurons (i.e., TPH 1) [56,57]. In the *raphe*, serotonin synthesis begins from the essential amino acid tryptophan provided by the diet [18,58] that crosses the BBB, accessing the brain; it is then converted into 5-hydroxytryptophan by TPH 2 [13,55]. Subsequently, 5-hydroxytryptophan undergoes decarboxylation driven by the L-aromatic acid decarboxylase, resulting in 5-hydroxytryptamine (i.e., 5-HT), namely serotonin [13,55] (Figure 3A).

Successively, serotonin is packed into synaptic vesicles thanks to the action of the vesicular monoamine transporter isoform 2 (VMAT2). Serotonin exerts its function both locally after release to specific postsynaptic targets, and broadly reaching distant brain regions [13,51,55]. Generally, neuronal communication occurs at synapses, where neurotransmitters are released in the synaptic cleft and act on specific postsynaptic receptors.

The termination of serotonin action on its receptors involves its reuptake and enzymatic degradation (Figure 3A). Extracellular serotonin is transported back into the neuron by the serotonin reuptake transporter (SERT) and metabolized by monoamine oxidase (MAO) enzymes, primarily MAO-A [13]. This process also occurs in surrounding glial cells, where MAO-A catalyzes the oxidative deamination of serotonin, producing 5-hydroxy-3-indolacetaldehyde (5-HIAL) further converted into 5-hydroxy-3-indolacetic acid (5-HIAA) by aldehyde dehydrogenase. The final product, 5-HIAA, represents an inactive metabolite that is ultimately eliminated [19,59].

**Figure 2 ijms-26-05508-f002:**
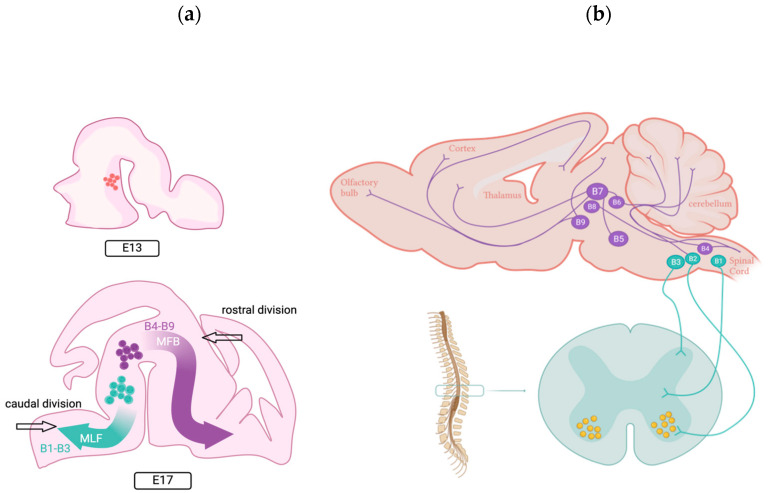
**Development of the serotonergic system**. (**a**) Schematic cartoons of rodent sagittal brain sections at embryonic days 13 (E13, (**top**)) and 17 (E17, (**bottom**)), showing the emergence and clustering of serotonergic neurons. By E17, serotonergic neurons are grouped into caudal (B1–B3, light green) and rostral (B4–B9, purple) clusters. Serotonergic fibers destined for the forebrain (purple arrow) travel along the median forebrain bundle (MFB) and originate from the dorsal and median *raphe*. Serotonergic fibers destined to the spinal cord (light-green arrow) arise from the B1–B3 clusters (red circles) and travel along the medial longitudinal fasciculus (MLF) [adapted from [60]. (**b**) Serotonin innervation in the central nervous system. (**Top**): schematic diagram showing the major rostral (purple) and caudal (green) serotonergic projections. (**Bottom**): serotonergic innervation of the spinal cord, with the *raphe pallidus* (B1) projecting to the ventral horn, the *raphe obscurus* (B2) innervating the intermediate zone, and the *raphe magnus* (B3) projecting to the dorsal horn of the spinal cord. Yellow circles indicate motor neurons [adapted with permission under the terms of the Creative Commons Attribution License, available at https://creativecommons.org/licenses/by/4.0/legalcode.en (accessed on 27 May 2025) (CC-BY License) from [60]. Created with BioRender.com.

Serotonin exerts its functions via fourteen classes of receptors belonging to seven families (5-HT_1_ to 5-HT_7_) according to their structural, functional, and pharmacological properties (Figure 3B). Out of these, members of six subfamilies (5-HT_1_, 5-HT_2_, 5-HT_4_, 5-HT_5_, 5-HT_6_, and 5-HT_7_) are G protein-coupled receptors (GPCRs), whereas members of the third family subtype (5-HT_3_) are ligand-gated ion channels [13,14,18]. The numerous serotonin receptors are largely and diversely distributed within the brain, acting differently in modulating the balance of neural excitability [13,14]. The 5-HT_1A_ receptor is expressed by all serotonergic neurons as a presynaptic inhibitory auto-receptor in the *raphe nuclei* whereas, postsynaptically, it is expressed on cholinergic neurons in the forebrain and in glutamatergic neurons in the neocortex and the hippocampus [61]. The 5-HT_1B_ receptor is located mostly on axon terminals [61] and it exerts its function either as an auto-receptor, preventing serotonin release, or as a heteroreceptor controlling the dispensation of diverse neurotransmitters on non-serotonergic neurons [61,62]. The 5-HT_2_ family members are predominantly 7-transmembrane-spanning (7-TMS) membrane proteins and are localized within the cerebral cortex. They act by inducing negative feedback via GABAergic and glutamatergic neurons that contrast with the firing of serotonergic neurons of the *dorsal raphe nuclei*. When activated, these receptors stimulate local GABAergic interneurons, increasing the release of GABA, which inhibits the firing of serotonergic neurons. This mechanism forms a negative feedback loop that helps to regulate serotonergic output [63]. The 5-HT_2B_ receptors in astrocytes are triggered by serotonin-specific reuptake inhibitors, such as fluoxetine, one of the major antidepressants [64]. The 5-HT_2B_ subfamily members might have a role in antidepressant treatment response and may represent a new target for tailored depression therapies [65].

The 5-HT_3_ receptors are ligand-gated ion channels, The expression pattern entails the spinal cord, the brainstem, and—to a lesser extent—the limbic system and the neocortex [13,61]. The 5-HT_3_ receptors are involved in inhibiting pyramidal neurons in the medial prefrontal cortex via both the activation of GABAergic interneurons [64] and through modulating the release of acetylcholine and dopamine [66]. The 5-HT_3_ receptors have a role in thermoregulation [61,67]. Moreover, they displays anxiogenic and emetogenic properties, since agonists can elicit unpleasant nausea and anxiety effects [66,68,69].

In addition, the 5-HT_4_ receptors are widely expressed within several tissues, and are involved in the slow excitatory response to serotonergic neurons [4,70,71,72]. Recent findings have focused on the peripheral and central effects of 5-HT_4_ in different disorders, including anxiety, addiction, gastroesophageal reflux, and irritable bowel syndrome [70,71,72]. Structurally distinct from other receptors, 5-HT_5_ members feature some pharmacological characteristics which overlap with those of 5-HT_1D_. Nevertheless, their clinical significance and their involvement in diseases are yet to be explored, in particular, whether they are implicated in the pathogenetic mechanism of epilepsy [72,73,74]. Regarding the 5-HT_6_ receptor, current data support its possible proconvulsive action [72,74]. Although 5-HT_6_ appears to be implicated in numerous neuropsychiatric diseases, the clinical significance is still lacking [72,75]. Structurally homologous to the 5-HT_1A_ receptor, 5-HT_7_ is, from one recent description, a member of the G protein-coupled receptor family, and it shares an identical second messenger (adenylyl cyclase) with the 5-HT_1A_ receptor, but with opposite activity: the first activates it, while the second blocks it [61]. Because of its wide distribution in the central nervous system, 5-HT_7_ receptors are involved in several pathophysiological mechanisms, such as mood and the circadian rhythm [72,76,77].

**Figure 3 ijms-26-05508-f003:**
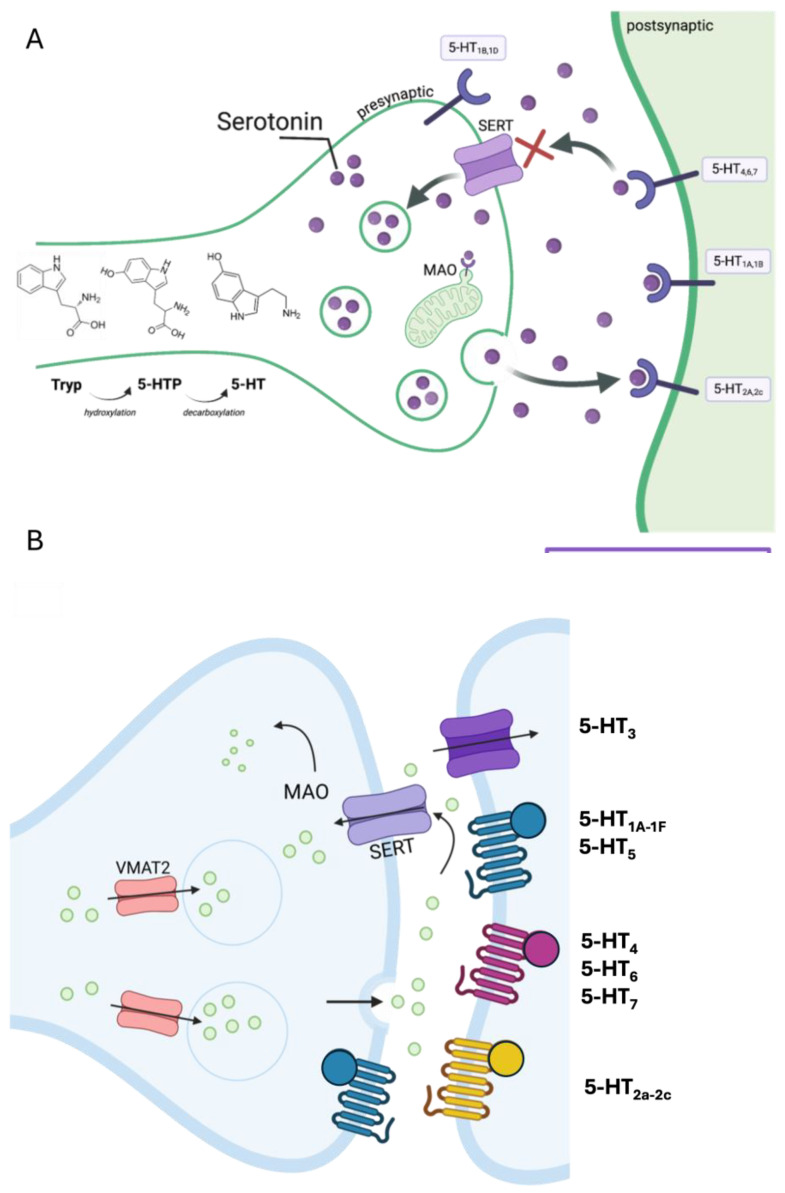
**The serotonergic system**. (**A**) Schematic cartoon of a serotonergic synapse with pre- and postsynaptic sites. The tryptophan (Tryp) introduced to the diet is hydroxylated into 5-hydroxytryptophan (5-HTP) by tryptophan 5-hydroxylase 2 (THP 2), and then decarboxylated into 5-hydroxytryptamine (5-HT) or serotonin by the aromatic L-amino acid decarboxylase enzyme (DDC). Once packed into the vesicles, serotonin is released into the synaptic cleft and binds postsynaptic receptors (i.e., 5-HT_1A_, 5-HT_1B_, 5-HT_1D_, and 5-HT_2A,C_). Serotonin is transported into the presynaptic terminal thanks to the action of the serotonin reuptake transporter (SERT) present in the plasma membrane. Here, 5-HT is broken down by MAO (monoamine oxidase) and, for future release, is repackaged into vesicles by isoform 2 of the vesicular monoamine transporter (VMAT2). Created with BioRender.com. (**B**) Schematic cartoon of the 5-HT receptor families. MAO: metabolic enzymes monoamine oxidase; SERT: serotonin reuptake transporter; VMAT2: vesicular monoamine transporter 2. [adapted with permission under the terms of the Creative Commons Attribution License, available at https://creativecommons.org/licenses/by/4.0/legalcode.en (accessed on 27 May 2025) (CC-BY License) from [78].

## 2. GABA and Serotonin as Main Players During Neurodevelopment

Neurodevelopment allows the formation and maturation of neuronal circuits in order to achieve the mature functions of the adult brain. Molecular, genetic, and environmental factors regulate neurodevelopmental processes [79,80]. This regulation is also heavily influenced by epigenetics, heritable changes in gene expression not directly explained by genomic information alone [79]. During cortical development, neural progenitor cells (NPGs; namely, cells already committed to a neural fate) migrate to their final location, differentiate, and establish synaptic contacts with their targets [3]. In mammals, the cerebral cortex develops from the rostral telencephalon through a process underpinning several connected events that generates a six-layer lamina, a complex structure responsible for motor, cognitive, and sensory functions [4,79,81]. Cerebral cortex development follows an “inside-out” sequence where pyramidal neurons that generate at earlier embryonic ages are localized closer to the ventricular zone (VZ), and pyramidal neurons generated at later embryonic ages migrate to the surface of the cortical plate [17,65]. It is critical that, through cell migration, pyramidal neurons reach their final target in the brain, as abnormal neuron localization and disrupted wiring lead to cortical malformations, seizures, and cognitive deficits [65,82,83]. Each cortical layer contains both pyramidal (glutamatergic) and non-pyramidal (GABAergic) neurons [1,2,3] that originate early during development respectively from the VZ and the medial ganglionic eminence (MGE) [2,9,65,84]. Excitatory neurons migrate radially, while inhibitory neurons migrate tangentially [85,86,87,88] (Figure 4).

GABA plays a fundamental role during neurodevelopment, regulating processes such as neuronal formation, migration, and differentiation [6]. GABA action through ionotropic GABA_A_ receptors is depolarizing, thanks to the higher expression of NKCC1 in developing neurons [6,81]. Apart from the key role of GABA in controlling brain development, a modulatory role is also played by serotonin [17]. The interactions between these contribute to shaping the various regions of the nervous system [89]. As mentioned earlier, the serotonin source in a developing brain is the clusters of neurons proliferating within the neuroepithelium and projecting to the cortical plate (see Figure 2). Although *raphe* serotonergic neurons originate during embryonic development—in rodents at embryonic (E) days 10–12—their maturation is prolonged in postnatal life (in rodents, postnatal (P) day 21) [90]. In detail, the progressive arrival of inhibitory and excitatory inputs on both GABA and serotonergic *raphe* neurons—from P4 to P21—is correlated with the refinement of electrophysiological firing properties, as is the appearance of the 5-HT_1A_ auto-receptors [91]. This ongoing maturation and refinement of the serotonergic signaling during postnatal development highlights the critical role of the serotonergic system in shaping the functional properties of the brain [90,92,93,94].

Since serotonergic neurons are already in contact with central neurons at early developmental stages [95], dysfunctions occurring in this time period might lead to long-term functional and structural variations implicated in the pathogenesis of a wide range of neurological and neuropsychiatric disorders [18,51,96,97,98]. In this process—occurring during CP plasticity windows—different fundamental steps need to happen to allow correct neuronal network development [99,100].

**Figure 4 ijms-26-05508-f004:**
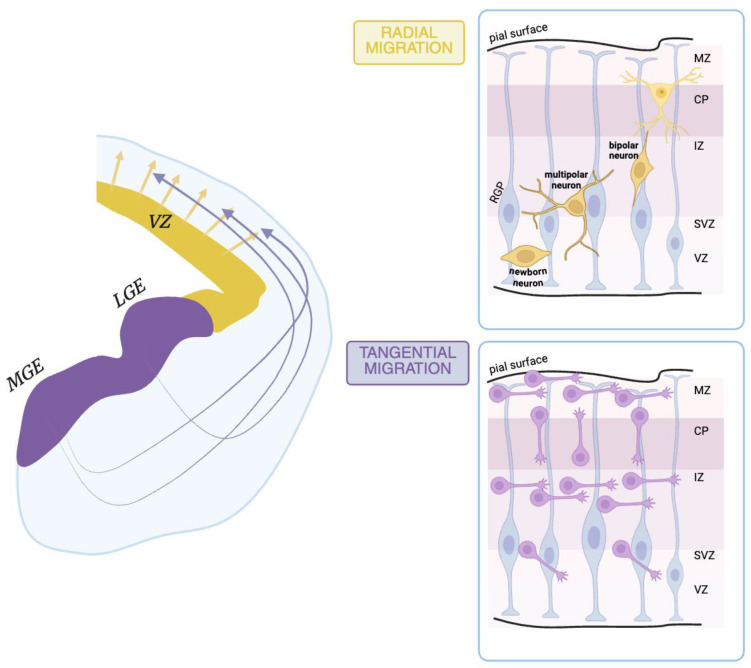
**Schematic representation of radial and tangential migration**. (**Left**): There are two primary types of neuronal migration: radial and tangential. In radial migration, pyramidal (glutamatergic) neurons travel perpendicularly to the ventricular surface along radial glial fibers (yellow). In contrast, during tangential migration, inhibitory (GABAergic) neurons follow paths that are parallel to the ventricular surface and perpendicular to the radial glial fibers (purple) [adapted with permission under the terms of the Creative Commons Attribution License, available at https://creativecommons.org/licenses/by/4.0/legalcode.en (accessed on 27 May 2025) (CC-BY License) from [101]. (**Right**): Specifically, excitatory pyramidal neurons are generated from glial progenitor cells (RGPs) in the VZ and migrate radially toward the cortical plate (CP), following the apical RGP process. MZ: marginal zone; CP: cortical plate; IZ: intermediate zone; SVZ: subventricular zone. Non-pyramidal inhibitory neurons, on the other hand, arise from the medial ganglionic eminence (MGE), migrate tangentially in streams, and later invade the cortex. Created with BioRender.com.

### 2.1. During Prenatal Life

During prenatal brain development, a homogenous embryonic neural plate morphologically shapes into a complex nervous system formed by several brain cell types, complexly connected [102]. These cell types in the adult brain originate from progenitor pools in distinct subregions of the developing brain, which are established early during the neural tube’s morphogenic patterning [1,2,103]. These mechanisms are driven by extrinsic and intrinsic signals as molecular cues and transcription factors, to accomplish fine-tuned regulation.

GABA plays a main role in regulating neuronal migration, synapse formation, and differentiation through phasic and tonic GABA_A_ neurotransmission [6,7,104].

During the first postnatal week—when synapses start growing—GABA_A_ receptors are subjected to a massive GABA concentration that spreads within the neuropil shaping synaptic currents; thus, GABA is picked up by specific transporters that contribute to its clearance from the extracellular space [105]. This transient GABA action is important for timing-based signaling, setting the temporal window for synaptic integration [106] and the synchronization of neuronal networks [107]. In regard to GABA polarity, given the higher intracellular Cl^−^ concentration driven by NKCC1 activity, GABA action on synaptic GABA_A_ receptors results in phasic GABA_A_ currents leading to membrane depolarization and neuronal excitation (i.e., action potentials firing). In the hippocampus, synaptic phasic GABA_A_ receptor-mediated neurotransmission is also depolarizing [12] and contributes—together with glutamatergic neurotransmission—to generating giant depolarizing potentials (GDPs) consisting of recurrent and suprathreshold depolarizations, synaptically evoked [6,31,108]. GDPs’ oscillatory activity synchronizes and wires immature neural circuits, enhancing hippocampal synaptic activity during neurodevelopment [31,109]. Conversely, extracellular GABA activates extra-synaptic GABA_A_ receptors, resulting in tonic GABA currents (Figure 1). Extracellular ambient GABA originates from (i) the astrocytes through a calcium-independent mechanism, not mediated by vesicles [63,110], (ii) leakage from the synaptic cleft [111,112,113], and (iii) reverse transport induced by mild depolarization or an increased cytosolic neurotransmitter concentration [104,114,115]. Therefore, extra-synaptic GABA_A_ receptors are continuously exposed to low concentrations of GABA [116] and generate the constant (tonic) current that underpins network excitability regulation and cell firing [117]. Both currents—phasic and tonic—play a pivotal role in neuronal migration [6,12,104]. Once neurons reach their final destinations, GABA_A_ receptor activation regulates differentiation [118] and integration into developing circuitry through new synaptic contacts [119] while also regulating the maturation of inhibitory connections [120].

Although GABA is a major player in brain development, serotonin also modulates neuronal migration, since serotonergic neurons emerge early in development (around E10–12 in mice and E12–15 in rats) and project to both subcortical and cortical regions, as well as the spinal cord [17].


**
*In the developing spinal cord*
**


During prenatal spinal cord development, serotonin is able to modulate the GABAergic phenotype of spinal neurons through descending inputs from the caudal division. In this context, serotonin acts as a delaying signal for GABAergic neuron maturation [121]. By using an organotypic culture, Allain and colleagues (2005) modified serotonin content and inputs within the embryonic mouse spinal cord and assessed the GABAergic phenotype [121]. Consistent with previous studies, the absence of all serotonin sources, achieved by the exportation of the *medulla* together with the use of p-chlorophenylalanine (drugs preventing serotonin synthesis by ectopic neuronal cells), led to a significant increase in the number of GABAergic neurons. Furthermore, the authors observed that, when the *medulla* was left intact while serotonin synthesis was blocked with p-chlorophenylalanine, the GABAergic cells developed synchronously along the spinal cord and the rostro–caudal axis, indicating the delaying effect of serotonin descending inputs, a process in which the 5-HT_1_ receptor family is involved [121].

Furthermore, a study by Martin and colleagues (2020) discusses the role of serotonin in the dysregulation of Cl^−^ homeostasis in motoneurons in a mouse model of amyotrophic lateral sclerosis (ALS) [122]. The authors demonstrated that serotonin—through its interaction with the GABAergic system—influences Cl^−^ homeostasis by interacting with KCC2. The interaction between serotonin and KCC2 may play a role in the early neurophysiological changes observed in ALS, particularly in familial cases with the SOD1G93A mutation, thereby contributing to disease’s progression [122].


**
*In the developing cerebellum*
**


In mankind, cerebellar cortex development begins in the first trimester of pregnancy and continues until 2 years of postnatal age [123]. In a pioneering experiment, Strahlendorf et al. (1989) demonstrated that serotonin modulates GABAergic control on Purkinje cells in two phases: initially, serotonin reduces GABA-mediated inhibition, followed by an enhancement of GABA responses. Interestingly, the dual effects of serotonin on GABAergic inhibition appear to depend on the intrinsic properties of Purkinje cells, likely associated with their firing rate [124].

More than 30 years later, Oostland and van Hooft (2011), using immunohistochemistry and electrophysiological recordings in organotypic slice cultures, observed that serotonergic input activates 5-HT_2_ and 5-HT_3_ receptors in the cerebellar cortex [125], and that reelin—a 5-HT_3_ receptor blocker—increased Purkinje cells’ dendritic complexity during early postnatal development, thereby suggesting the key role of serotonin in the growth and stability of cortical cerebellar synapses [126]. Moreover, the authors showed that serotonin had a high receptor distribution both in the developing and in the adult cerebellum, and a low density of SERT during the prenatal period [127,128]. They also noticed that the 5-HT_1_ receptor exerted a biphasic current in the Purkinje cells and an inward current in the granule cells through activation of the G protein-coupled mechanism [126].

Apart from the development of Purkinje cells, experiments have also unraveled the role of GABA and serotonin in Lugaro cells, namely, inhibitory interneurons in the cerebellar granular layer [129]. Diedudonnè and co-workers (2000) used patch clamp recordings at P16–25 and found that Lugaro cells influence the inhibitory input from Golgi cells to large granule cell populations by delivering synchronized inhibition to multiple Golgi cells [130]. Moreover, in a subsequent study, it was shown that 5-HT_1A_ receptors are transiently expressed during the initial two postnatal weeks, whereas 5-HT_2A_ receptors persist on granule cells until 10 weeks of age and are fundamental for Purkinje cell synaptic plasticity in the cerebellar cortex [127].

Taken together, these studies highlight the key roles played by GABA and serotonin in the developing spinal cord and cerebellum.

### 2.2. During Postnatal Life

Once embryonic development comes to an end, synaptogenesis and spontaneous neuronal activity refine the immature neuronal networks in the early postnatal life [96,97]. Later, during postnatal life, sensory experience continues to sculpt immature networks during CP windows of plasticity. In particular, CP plasticity refers to temporal windows during neurodevelopment when the nervous system is particularly sensitive to experience and environmental stimuli, resulting in proper maturation of brain functions within CP. In other words, during CP, the brain is easily shaped by sensory input and is experience-dependent, and this has long-lasting effects on brain structure and function. Moreover, these time windows of plasticity are “critical” because an absent or altered experience will have a devastating effect on brain structure and function. For example, a unilateral eye cataract during early postnatal development in children results in amblyopia (i.e., lazy eye) [131,132,133]. Maldevelopment during CPs has been linked to a number of neurodevelopmental neuropsychiatric disorders [134,135], and proper sensory experiences are essential during CPs to achieve proper neurodevelopment. After CP plasticity closure, plasticity decreases or is limited, rendering the modification of neural circuits more difficult and less effective in adulthood [99,100,131,132].

In humans, the motor cortex undergoes a critical period of plasticity between the ages of 6 and 8, during which children naturally acquire and refine complex motor skills, such as walking and grasping, thanks to the brain’s heightened adaptability [136]. The auditory cortex has its plasticity window from about 6 months to 4 years of age, while the visual cortex enters its critical phase between 3 and 5 years [137]. The hippocampus experiences its peak plasticity between 2 and 4 years, although its development continues into adolescence [138].

Regarding the role played by serotonin during postnatal life, pioneering studies showed that serotonin accelerated synaptogenesis and axon myelination when administered to organotypic cultures. This effect was probably due to the direct action on neurons via metabotropic and ionotropic receptors [104,139]. Moreover, during postnatal life, serotonin interacts with GABA. This is important, for example, for the control of the mossy and climbing fibers of the cerebellum via GABA inhibition with a decrease of phasic GABA_A_ currents [140]. Another example of this interaction comes from studies of 5-HT_3_ receptor activation in the presynaptic GABA terminals of the CA3 region that bring an increase in the GABA release [141].


**
*In somatosensory cortex development*
**


GABA and serotonin interact during sensory cortex development [142]. The distribution of cortical neurons within the somatosensory cortex of rodents is mapped and already established during the first week of life, as thalamocortical axons and cortical sensory regions finalize their positions [143]. The primary somatosensory cortex (S1) features distinctive structures known as barrels. The term was introduced by Woolsey and Van der Loos in 1970, to describe cylindrical structures observed in S1 of rodents, resembling cylindrical “barrels” when stained, and analyzed in histological cross-sections [144]. These structures are highly organized clusters of layer IV granule cells and thalamocortical axons that mirror the spatial arrangement of whiskers and sensory hairs on the snout [145]. Barrel formation occurs during early development through the interaction of thalamocortical axons and granule cells. Sensory input during the perinatal period is critical for their formation, as disruptions to the sensory periphery during CP prevents proper thalamocortical axon patterning, barrel organization [145], accompanied by the reduced maturation of the GABAergic system with a decrease in the GABA/AMPA current ratio and depolarized equilibrium potential for Cl (E_Cl_), and reduced expression of the KCC2 cotransporter [146].

Conversely, decreased serotonin levels cause the delayed maturation of the barrel cortex, as shown in works using antidepressant administration in an MAO-A-deficient mice model and a juvenile SERT knockout model (SERT^−/−^) [99,147,148]. In MAO-A-deficient mice pups, serotonin levels were higher in the first postnatal weeks in comparison to old mice [99,147]. Furthermore, administration of serotonin-specific reuptake inhibitors (SSRIs) elicited an increase in serotonin levels, causing a suppression of the spontaneous cortical activity in S1 [149]. Pharmacological depletion of serotonin with 5,7-dihydroxytryptamine (5,7-DHT) resulted in a prolongation in the maturation of the barrels [150]. In S1, in nearly all GABAergic parvalbumin (PV)- and somatostatin (SST)-negative interneurons, the activation of 5-HT_3A_ receptors induces a strong depolarization, followed by burst firing. This process suggests that serotonin might alter cortical activity by quickly enhancing GABAergic synaptic transmission depending on different behavioral states [151].

One interesting study investigated how the serotonin-dependent modulation of GABAergic transmission during somatosensory cortical development impacts behavior. Naskar and co-workers (2019) showed that synaptic GABA transmission is time-locked to a serotonin upsurge and to an early form of social behavior, namely, huddling between littermates. When citalopram—a SSRI—was administered to mice pups, the authors found a shift in the appearance of the GABA transmission appearance due to increased serotonin levels from the huddling behavior earlier [152]. 

Taken together, these studies show that serotonin modulates GABA action during neurodevelopment in the somatosensory cortex.


**
*In visual system development*
**


There are two distinct phases in visual system development, maturation, and plasticity, temporally divided by the eyes opening: (i) an experience-independent phase, driven by molecular cues or a spontaneous pattern of electrical activity, and (ii) an experience-dependent phase, driven by direct patterned visual experience [100,131,132,153]. In the second phase of postnatal visual development, the visual system undergoes to sensory-driven experience-dependent maturation during a CP window characterized by a high level of plasticity [154].

Different studies using in vivo and in vitro electrophysiology have shed light on the role of visual experience in visual system maturation and plasticity and have described the existence of CP plasticity windows. For example, Hubel and Wiesel performed pioneering experiments using in vivo electrophysiological single-unit recordings in anesthetized cats. In normal conditions, in both young and adult animals, Hubel and Wiesel classified neurons in the primary visual cortex (V1) into seven different classes of ocular dominance (i.e., the preference of the visual cortical neurons to modulate their firing frequency upon stimulation of either the ipsilateral or contralateral eye). Strikingly, if, in young animals, one of the two eyes was deprived of visual experience for few days (by suturing the eyelids), the authors found a shift in the ocular dominance distribution (i.e., ocular dominance plasticity), with neurons not responding anymore to the eye deprived of vision, but to the open eye that had become the dominant eye. This shift was not observed when visual deprivation was performed in adult animals [155], highlighting the existence of sensitive critical windows during postnatal development. Importantly, the drastic shift in ocular dominance distribution was correlated with drastic negative effects in the normal maturation of visual functions (low visual acuity), and these effects persisted in adult life.

The mechanism of ocular dominance plasticity relies, on an anatomical level, on the retraction of the thalamo-cortical axons serving the deprived eye (in cats) [96]. In mice, an expansion of the thalamo-cortical axons, guided by the open eye, was observed [156]. In vivo experience-dependent modifications during CPs are phenocopied in in vitro experiments of long-term potentiation (LTP), following stimulation of the white matter (WM). WM-LTP in visual cortical slices can be induced only within CPs, and not after their closure [157].The pioneering experiments of Hubel and Wiesel, together with the extensive scientific literature describing in vivo and in vitro evidence, shed light on the existence of postnatal CP plasticity windows for sensory maturation in the visual system. They further describe CP windows for other sensory modalities and the development of motor skills [131,132,158].

GABA, via GABA_A_ receptors, plays a major role in visual cortex CP plasticity control. In fact, hyperpolarizing GABA controls both the opening and the closure of a CP [132,137,159], while early depolarizing GABA (before the eyes open) has a long-lasting effect in controlling CP closure [85]. Since visual sensory maturation occurs during CPs, it is broadly accepted that visual cortical plasticity and the maturation of visual functions are two correlated and casual processes. In fact, manipulations targeting the GABA system that alter CP onset, closure, or duration also impact visual maturation (i.e., visual acuity) [157,160,161,162]. However, one study demonstrated that, under certain experimental manipulation—for example, through interfering with the regulation of CP plasticity with early depolarizing GABA—the two processes (plasticity and the maturation of visual functions) could be untangled.

While GABA exerts major control over visual cortical plasticity, serotonin plays a modulatory role. Serotonin depletion in vivo with 5,7-DHT shifted ocular dominance distribution in monocularly deprived young kittens [163]. Gu and coauthors found comparable results in ocular dominance plasticity when using 5-HT_1_ and 5-HT_2_ receptor antagonists (for example, mesulergine) [163,164]. Serotonin modulates in vitro LTP in visual cortical slices. Edagawa and colleagues observed an inhibition of LTP through endogenous serotonin activation in layers IV and II–III in the visual cortex of Wistar rats [165]. Maya Vetencourt and co-workers (2008) used the antidepressant fluoxetine in vitro and in vivo in adult rodents (i.e., when the CP had already closed) and found (i) an ocular dominance plasticity shift and recovery of visual functions in the adult age after long-term monocular deprivation, and (ii) WM-LTP induction, together with a reduction in GABA concentrations in the primary visual cortex V1 [166]. Apart from pharmacological manipulations, dark rearing (total visual deprivation paradigm) or environmental enrichment (increased sensory and motor stimulation) protocols prolong and accelerate, respectively, CP opening and closure [132,160,167,168,169,170]. Altogether, this evidence indicates that GABA and serotonin regulate CP plasticity in the visual system.

TRK_B_ receptors, expressed broadly in hippocampal and cortical excitatory and inhibitory neurons, play a pivotal role in synaptic plasticity [171]. Parvalbumin-positive (PV^+^) interneurons are fast-spiking cells that provide GABA-mediated inhibition to pyramidal neurons synapsing in the perisomatic subcellular compartment. Each PV^+^ neuron connects with hundreds of pyramidal neurons, exerting widespread control over activity and coordinating gamma oscillations essential for network synchronization [172]. The postnatal maturation of PV^+^ neurons is closely associated to the closure of CP plasticity [27,173]. Notably, research on rodents has demonstrated that pharmacological treatments such as serotonin-targeted antidepressants reduced intracortical inhibition and induced juvenile-like plasticity in the adult brain through the activation of TRK_B_ receptors [166,174]. Thus, drugs targeting the serotonergic system—that modulates synaptic plasticity via TRK_B_ receptors—can be considered a valuable therapeutic target to re-open CP plasticity windows into adulthood for neuropsychiatric interventions [175].


**
*In cerebellum development*
**


As mentioned before, the development of the cerebellum persists until 2 years of postnatal age [123]. Saitow and colleagues used patch clamp recordings in P15 Wistar rats and found that serotonin modulates deep cerebellar nuclei activity by presynaptically inhibiting GABA release [176]. Since the activity of neurons in the cerebellar nuclei is controlled not only by excitatory inputs (mossy and climbing fibers) but also by GABA inputs from Purkinje cells, the latter can be modulated by serotonin. As shown in in vitro studies, serotonin decreases the amplitude of GABA_A_ currents, with a stronger effect in the second postnatal week than in the third one [140]. Serotonin also contributes to the cerebellum circuitry development in postnatal life. In particular, in rats from P10 to P21, it was found that serotonergic fibers extended to the Purkinje cell with consistent arborization in the cerebellar cortex [177].

In summary, the studies reviewed so far highlight the fundamental roles of GABA and serotonin in controlling and modulating pre- and postnatal development and plasticity.

### 2.3. During Adult Life

Brain plasticity is not confined only to pre- and postnatal life [178], but it is present—to a lesser extent—also in adult life [179,180,181,182]. For example, adult neurogenesis refers to the production of new neurons in specific neurogenic niches during adulthood [183,184], namely, within the forebrain subventricular zone (SVZ) [185,186] and the hippocampal dentate gyrus (DG) [170,187]. In these adult brain niches, neurons undergo a similar process to that which occurs during early brain development [181]. In particular, the SVZ produces dopaminergic, GABAergic, and glutamatergic neurons that migrate along the rostral migratory stream to integrate into the olfactory bulb [188,189]. Meanwhile, the DG generates granular neurons involved in memory, learning, and cognitive flexibility [190,191,192].

As during early development, GABA and serotonin play main and distinct roles in adult neurogenesis [188,193]. Serotonin influences neural stem cell proliferation in SVZ via the activation of 5-HT_2C_ receptors, significantly enhancing the proliferation of B1-type cells [194]. In addition, chronic administration of the SSRI fluoxetine further enhances neurogenesis by increasing BDNF expression, particularly in the medial habenula [195]. GABAergic signaling, on the other hand, regulates neuroblast proliferation within the SVZ through feedback inhibition. Neuroblasts release GABA, which activates GABA_A_ receptors on progenitor cells, thereby ensuring proper timing and functionality in olfactory circuit refinement. This mechanism is critical for maintaining the balance between neuronal production and integration into existing neural circuits [196].


**
*In the hippocampus*
**


Concerning serotonin, Malberg and colleagues (2000) assessed, using immunohistochemistry, the consequences of a chronic antidepressant treatment on hippocampal neurogenesis in adult male rats [197]. After 4 weeks of subsequent administration of bromodeoxyuridine (BrdU, labeled as proliferating cells in the DNA-replicating phase), the authors assessed cells in the dentate gyrus of the hippocampal in terms of differentiation, proliferation, and survival. The number of BrdU-positive cells increased, suggesting enhanced neurogenesis after the treatment [197]. Malberg and colleagues used fluoxetine in female rats, seeing comparable results [198]. The next question to be answered was whether the increased adult neurogenesis induced through the fluoxetine administration was necessary for the antidepressant action of the SSRIs. Santarelli and colleagues explore this gap by carrying out research on 5-HT_1A_ knockout mice [199]. They first confirmed that fluoxetine increased adult neurogenesis, as seen in the previous study by Malberg and collaborators. Next, they explored the effect on 5-HT_1A_ knockout mice. Interestingly, 5-HT_1A_ knockout mice were unresponsive both to the antidepressant fluoxetine and to the neurogenesis effect, demonstrating that 5-HT_1A_ receptors are necessary to mediate the effects of fluoxetine treatment on adult neurogenesis and the depressive-like phenotype. Fluoxetine failed to induce the antidepressant phenotype when hippocampal neurogenesis was disrupted with X-ray radiation [199], confirming the necessity of adult neurogenesis to mediate the antidepressant effects. The causal links among fluoxetine, neurogenesis, and depression have been similarly described in other models of depression [200,201].

Despite it being widely known that fluoxetine—as an SSRI drug—elicits an increase in free serotonin [202], a recent study by Olivas-Cano and co-workers (2023) showed that fluoxetine increases adult neurogenesis via its effect as a selective 5-HT_3_ antagonist, thereby extending to the 5-HT_3_ receptor its mechanisms of action [202].

Serotonin modulates plasticity during neurodevelopment and, in adult age, also via its interaction with GABA [203]. In fact, serotonin regulates GABAergic transmission at the presynaptic and postsynaptic level [204]. At the presynaptic level, serotonin prevents GABA release through the 5-HT_1A_ and 5-HT_1B_ receptors and promotes it through the 5-HT_2_ and 5-HT_3_ receptors. At the postsynaptic level, serotonin modulates GABAergic transmission via several receptors, as described, in the hippocampus, in the thalamus, or in the prefrontal cortex [203,205,206,207,208].

## 3. Psychedelics to Re-Open Critical Period Windows for Therapeutic Interventions in Neuropsychiatric Disorders

The pieces of evidence described so far highlight the roles played by GABA and serotonin during pre- and postnatal neurodevelopment. Indeed, a plethora of evidence collected in the last decades shows that these neurotransmitters—individually and cooperatively—exert fundamental control and have a modulating effect both during neurodevelopment and adult plasticity [209,210]. Since CP closure restrains brain plasticity at adult ages [131,132,211], the question that arises is whether a modulation of the serotonin–GABA interplay—already physiologically in motion during neurodevelopment—can be used as an asset to identify new potential strategies to re-open CP plasticity in adulthood for the treatment of neuropsychiatric disorders [29,212]. To answer this question, different research groups recently explored the preclinical therapeutic potential of selected compounds (see Table 1).

Ly and co-workers (2018) suggested that the plasticity-promoting effects of psychedelics are comparable to those elicited by brain-derived neurotrophic factor (BDNF). In their experiments, the authors treated cortical cultures for 24 h with representative compounds, i.e., lysergic acid diethylamide (LSD), N, N-dimethyltryptamine (DMT), and DOI, that belong to the three main classes of psychedelics: ergoline, tryptamine, and amphetamine, respectively. These drugs exert their psychedelic effects by acting primarily on 5-HT_2A_ receptors [213,214,215]. The experiments of Ly and co-workers (2018) showed that serotonergic psychedelics are effective in intensifying spinogenesis and neuritogenesis in vitro and in vivo. According to these findings, it is conceivable that serotonergic psychedelics could induce both structural and functional changes in cortical neurons. Finally, the authors introduced the term “*psychoplastogen*” to classify these drugs and reflect their ability to promote neural plasticity [216]. These substances, particularly those active in the prefrontal cortex, show promising potential as antidepressants and anxiolytics and as templates for safer therapeutic alternatives, supporting their clinical value in the treatment of neuropsychiatric disorders [217].

Nardou and colleagues (2019) found out that 3,4-methylenedioxymethamphetamine (MDMA) in adult mice (P96) re-opened CP plasticity for social reward learning by enhancing oxytocin-induced long-term depression (LTD) in the *nucleus accumbens* [218]. Oxytocin induces LTD in excitatory synapses and, with serotonin, plays a crucial role in social reward, a mechanism that diminishes with age [219]. The authors administered intraperitoneally a single MDMA dose and observed increased scores in the social preference task and oxytocin LTD in the *nucleus accumbens* for 48 h following the treatment. These authors provided the first evidence that the CP plasticity for social reward learning is achieved through metaplastic changes in oxytocin-dependent LTD via the activation of oxytocin receptors in the *nucleus accumbens*, linking in this way MDMA prosocial effects with behavioral and synaptic outcomes. This finding is consistent with previous reports that MDMA binds SERT and determines the oxytocin release through the activation of 5-HT_4_ receptors expressed on oxytocin neurons [220,221,222]. Nardou and colleagues (2019) speculated that the psychedelic compound MDMA could be a promising new pharmacological tool for therapeutic interventions [218].

Next, Revenga and colleagues (2021) reported a post-acute increase in the density of immature and transitional dendritic spines following a single administration of the 5-HT_2A/2C_ agonist phenethylamine psychedelic 2,5-dimethoxy-4-iodoamphetamine (DOI) [223]. In the frontal cortex of vehicle-treated mice, spine density was lower than in 5-HT_2A+/+_ controls. Interestingly, these spine-related changes were not detected in 5-HT2A receptor knockout mice, suggesting a potential association between the presence of 5-HT2A receptors and DOI-related effects on dendritic spine arrangement. Notably, DOI was also associated with altered synaptic plasticity in pyramidal neurons, in particular within the layers II/III of the frontal cortex, since the remarkably higher LTP magnitude was revealed in DOI-treated compared to vehicle-treated mice [223]. The authors demonstrated the critical role of the 5-HT_2A_ receptor and opened up the potential of psychopharmacological therapies.

Desouza and colleagues (2021) observed an increase in the expression of several neuronal plasticity-associated genes induced by the 5-HT_2A_ receptor agonist and hallucinogenic DOI. Their findings highlighted that serotonergic psychedelic agents recruit the nuclear transcription factor CREB in order to induce transcriptional mechanisms, eventually resulting in plastic changes [224]. Furthermore, Ly and colleagues (2021) observed that a transient stimulation (mins or hours) with psychoplastogens, such as ketamine and LSD, triggered a phase of persisting neuronal growth that continued after drug withdrawal. LSD is a drug that induces altered perceptions and reduces the integrity of the brain networks, including increasing amygdala reactivity. On the other side, LSD increases thalamocortical activity involving the primary visual cortex and other several brain areas, thus leading to the hallucinations underpinning the subjective mystical experience [214]. These findings suggest that a transient administration might lead to enduring changes in neural circuitry, minimizing potential side effects from chronic daily use, such as those that have plagued patients using traditional antidepressants [225].

In a recent study, Davoudian and colleagues (2023) mapped the expression of the immediate early gene *c-Fos* following the administration of ketamine and psilocybin [226]. *c-Fos* is known to mediate key steps in synaptic potentiation and structural plasticity, providing a window into plasticity mechanisms [227]. Ketamine seems to act via AMPA and NMDA receptors, prompting the downstream production of BDNF and the activation of the mTOR signaling pathway [228]. The neurobiological mechanisms of psilocybin, despite it being a well-known 5-HT_2A_ receptor agonist, are not fully understood [229]. The authors used whole-brain serial two-photon microscopy and observed that ketamine and psilocybin elicited an acute increase in *c-Fos* expression in the primary visual cortex, amygdala, and *locus coeruleus*. The study suggests that NMDA receptor distribution predicts the *c-Fos* expression patterns evoked by both ketamine and psilocybin, particularly in the neocortex, pointing to the role of the glutamatergic system in mediating plasticity [226]. Another fairly recent study revealed that psilocybin-induced synaptic plasticity in the rat brain may be more pronounced in the prefrontal cortex compared to the hippocampus [230]; moreover, the author showed that neuroplastic gene expression, with corresponding variations in protein levels, might be dose-dependent [230].

Singleton and co-workers (2022) combined magnetic resonance imaging (MRI) and positron emission tomography (PET) to explore the effect of LSD and psilocybin on brain activity in mankind. Their findings revealed that 5-HT_2A_ receptors’ spatial pattern expression affects the energy landscape, influencing the energy required for shifting between different brain states, and this significantly more so than the other receptor distributions examined. This suggests that the 5-HT_2A_ receptor network may shape the impact of any neuropharmacological strategy for human brain dynamic activity [231].

An intriguing research question drove the work of Ornelas and colleagues (2022), who asked whether we could take advantage of psychedelic-induced neural plasticity to enhance cognition. In particular, they studied the relationship between learning, memory, aging, and neural plasticity by using the psychedelic LSD [214]. They observed that LSD pre-treatment is sufficient (i) to increase the expression of plasticity markers in human brain organoids, (ii) to enhance novelty preference in rats during behavioral tasks, and (iii) to facilitate visual memory consolidation and recall in humans [232]. In their findings, they also extend to humans the concept that psychedelic-induced structural plasticity encompasses the mTOR pathway, as described by Ly and colleagues (2018) [217,232].

It is noteworthy that growing evidence underlines the involvement of the GABAergic system as well [233,234]. The neurophysiological effects of psychedelics have been widely studied [233,234], particularly in the prefrontal cortex, where 5-HT_2A_ receptors are primarily postsynaptic and highly concentrated on the apical dendrites of deep-layer pyramidal neurons [235,236]. This distribution suggests that psychedelics, similar to serotonin, could enhance dendritic excitability and trigger excitatory postsynaptic potentials [237]. However, the in vivo effects of psychedelics are likely more intricate due to the diversity of cortical microcircuits, which include various subtypes of GABAergic neuron and different serotonin receptors [238,239], resulting in heterogeneous responses [240].

Reflecting this complexity, the systemic administration of DOI in rats elicited mixed effects on neuronal firing rates in the cortical neuron population [241]. Here, the authors investigated the effects of DOI (5HT_2A/C_ agonist), MK801 (NMDA antagonist), and amphetamine on the neuronal activity of freely moving rats. In particular, they focused on two subregions of the prefrontal cortex, namely, the orbitofrontal and the anterior cingulate cortices, whose role in schizophrenia is widely described [216,242]. Interestingly, following DOI treatment, the power of the gamma oscillations and the activity of the neuronal population were decreased. Conversely, NMDA administration elicited opposite effects. Unlike the disinhibition of cortical populations, this disruption in spiking and oscillatory activity networks may represent a shared pathway through which various molecular factors contribute to dysfunctional behavior and psychosis [241]. These results indicate that large-scale cortical activity analyses, such as examining population disruptions or spike–field interactions, could offer a novel electrophysiological framework to deepen the pathophysiology mechanisms in schizophrenia.

Indeed, reductions in glutamate release following psilocybin use are thought to result from the stimulation of 5-HT receptors on GABAergic interneurons [243,244]. Additionally, increased GABA levels have been observed in specific brain regions after use of psilocybin [245] or ayahuasca [246]. Dopamine signaling is also modulated by major psychedelic compounds, both indirectly through GABAergic or serotonergic mechanisms, or directly, with some speculation that psychedelics are converted into dopamine upon ingestion [247,248,249]. Ayahuasca, a psychoactive brew utilized for ritualistic, spiritual, and therapeutic purposes in Amazonian countries, contains DMT as the main psychotropic compound, and this is made orally active by β-carbolines, which inhibit monoamine oxidase to prevent its rapid breakdown [77,250]. Furthermore, DMT seems to influence neurotransmission by stimulating sigma-1 receptors, regulating glutamatergic, GABAergic, dopaminergic, and noradrenergic pathways through both direct and indirect actions [251].

The recent findings described in this review have begun to unravel the cellular and molecular mechanisms underlying the potential therapeutic effects of psychedelic drugs targeting the serotonin–GABA neurotransmission for re-opening CP plasticity windows in the adult brain for the therapeutic treatment of neuropsychiatric disorders. In particular, several psychedelics, such as psilocybin and LSD, as previously mentioned, primarily act as agonists on the 5-HT2A serotonin receptors, which are abundantly expressed in cortical pyramidal neurons. This serotonergic activation indirectly modulates the GABAergic system, particularly by reducing inhibitory tone through the disinhibition of interneurons. This interplay can facilitate cortical excitation and promote neural plasticity. Taken together, these studies provide evidence that these effects are associated with an ability to stimulate neurogenesis and neuroplasticity [217,252,253,254], shaping a new and intriguing milieu for investigating psychoplastogenic arrangements (Table 1).

**Table 1 ijms-26-05508-t001:** **Comparative summary of psychedelic compounds and their neuroplasticity-related effects.** This table highlights key studies investigating the effects of serotonergic psychedelics on neuroplasticity and behavior. It includes compounds tested (e.g., LSD, psilocybin, and ketamine), experimental models, underlying molecular mechanisms, observed effects on plasticity, and potential therapeutic implications.

Drug	Model	Mechanism	Effect	Relevance	**References**
MDMA	Adult mice (P96)	Via SERT and 5-HT4 receptors	Activation of oxytocin neurons restores oxytocin; long-term depression in the nucleus accumbens 48 h after a single administration, resulting in an improved score on the social preference task	Social reward learning	**Nardou et al. (2019) [218]**
DOI (5-HT_2A/2C_ agonist)	129S6/SvEv mice	Via 5-HT2A receptors	Increased expression of genes related to morphogenesis, neuron projection, and synapse structure; facilitated fear extinction	Schizophrenia, depression, hyperactivity disorder, depression, anxiety, and stressor-related disorders	**Revenga et al. (2021) [223]**
LSD, DMT, DOI	Sprague Dawley rats	TrkB, mTOR, and 5-HT_2A_ signaling	Increased dendritic spine density and enhanced neuronal excitability in the cortex, higher spontaneous excitatory postsynaptic current amplitude, and frequency in prefrontal cortical neurons	Psychoplastogens as potential new fast-acting antidepressants, and anxiolytic compounds	**Ly et al. (2018) [217]**
DOI	Male Sprague Dawley rats and (5-HT2A−/−) mice, 129S6/SvEv background, CREB-deficient mouse line,	5-HT_2A_ receptor via recruitment of CREB			**Desouza et al. (2021) [224]**
LSD, ketamine	Cortical cultures from Sprague Dawley rat	AMPA receptor and mTOR activation	Growth of cortical neurons, dendritogenesis, spinogenesis, and synaptogenesis	Implications for central nervous system drug development and neurotherapeutics	**Ly et al. (2021) [225]**
Ketamine, psilocybin	Eight-week-old C57BL/6J mice				**Davoudian et al. (2023) [226]**
Psilocybin	Sprague Dawley rats (7–9 weeks)	HT_2A_ receptor (PFC), 5-HT_1A_ (HIP)	Higher expression of genes related to neuroplasticity, and rapid regulation of plasticity-related genes in the prefrontal cortex and the hippocampus in a dose-dependent manner.	Further characterization of both acute and long-term molecular events induced by psilocybin	**Jefsen et al. (2021) [230]**
LSD, psilocybin	humans			Insights into pharmacological modulation of brain function	**Singleton et al. (2022) [231]**
LSD	Brain organoids, rats, humans	mTOR pathway	Increased plasticity markers in human brain organoids, enhanced novelty preference in rats, and improved visual memory consolidation and recall in humans	Clarification of the antidepressant and anxiolytic effects of serotonergic psychedelics; the possibility of alleviating and counteracting the cognitive deficits associated with natural or pathological aging	**Ornelas et al. (2022) [232]**
DOI, amphetamine, MK801 (NMDA antagonist)	Male Sprague Dawley rats		Systemic administration of 2,5-dimethoxy-4-iodophenyl-2-aminopropane in rats elicited mixed effects on neuronal firing rates in the medial frontal cortex	Schizophrenia	**Wood et al. (2012) [241]**

### 3.1. Clinical Trials and the Potential of Psychedelic Therapy

Remarkably, clinical research has confirmed the effectiveness of psylocibin in reducing depressive symptoms, prompting the Food and Drug Administration (FDA) to designate it as a “Breakthrough Therapy” for treatment-resistant depression in 2018 and for major depressive disorder in 2019 [255]. Studies emphasize its potential to produce sustained mood improvements, particularly in patients who do not respond to conventional treatments, especially as, unlike traditional psychiatric drugs, psychedelics carry a low risk of dependence [255,256].

The regulatory framework remains stringent, requiring approvals from agencies such as the FDA and the Drug Enforcement Administration (DEA) in the USA and the International Narcotics Control Board (INCB) worldwide [257]. Despite these barriers, several ongoing clinical trials continue to investigate the therapeutic potential of psychedelics, presenting promising alternatives for psychiatric and neurological disorders [257,258,259].

In future perspectives, a Phase 3 clinical trial (*NCT06308653*) is currently assessing the efficacy, safety, and tolerability of a 25 mg psilocybin dose in adults with major depressive disorder (MDD). This study will measure symptom changes from a baseline to 43 days post-administration, observing the one-year follow-up effects. Moreover, another Phase 3 clinical trial (*NCT05711940*) is exploring the impact of psylocibin administrations in 568 participants with treatment-resistant depression (TRD)—randomly receiving 25 mg, 10 mg, or 1 mg—alongside psychological support.

Similarly for Obsessive-Compulsive Disorder (OCD), there is a Phase 2 open-label trial (*NCT04882839*) aimed at exploring the feasibility of psilocybin treatment for severe conditions. The main purpose is to investigate new alternatives to continuously benefit patients even when conventional therapies are not working. Furthermore, a Phase 2 interventional study (*NCT06455293*) is currently recruiting 60 participants with Parkinson’s disease to evaluate the improvement of motor and non-motor symptoms through treatment with psilocybin. In this latter randomized controlled trial, participants who meet the depression criteria will undergo two sessions of psylocibin administration, side-by-side psychotherapeutic support, and investigations into other biomarkers through blood analyses and neuroimaging to constantly monitor quality of life.

In the treatment of addiction, ibogaine hydrochloride is a psychoactive alkaloid already known to mitigate hunger and thirst [260,261]. It is currently under investigation in two Phase 2 trials: *NCT04003948*, which examines its role in methadone detoxification, and *NCT03380728*, a completed double-blind placebo-controlled study assessing its effectiveness against alcoholism.

Although they face regulatory challenges, these clinical trials aim to provide significant evidence to support the integration of psychedelics into mainstream medicine, particularly for those conditions where existing therapies have limited success—thus, in this way, improving patients’ quality of life.

### 3.2. Concluding Remarks

In summary, the neurotransmitters serotonin and GABA play major roles in neurodevelopment and brain plasticity. Recent findings show that psychedelic substances—i.e., MDMA, DOI, and LSD—can re-open CP windows of plasticity in the adult brain. These compounds stimulate neurogenesis and synaptic remodeling, often through serotonin receptor activation—particularly 5-HT_2A_, which plays a significant role in neuroplasticity. Studies show that psychedelics may enhance dendritic spine formation and improve cognitive functions by engaging with the mTOR and CREB pathways. The findings reviewed here suggest that psychedelic drugs could serve as promising therapeutic treatments for neuropsychiatric disorders in the near future.

## Figures and Tables

**Figure 1 ijms-26-05508-f001:**
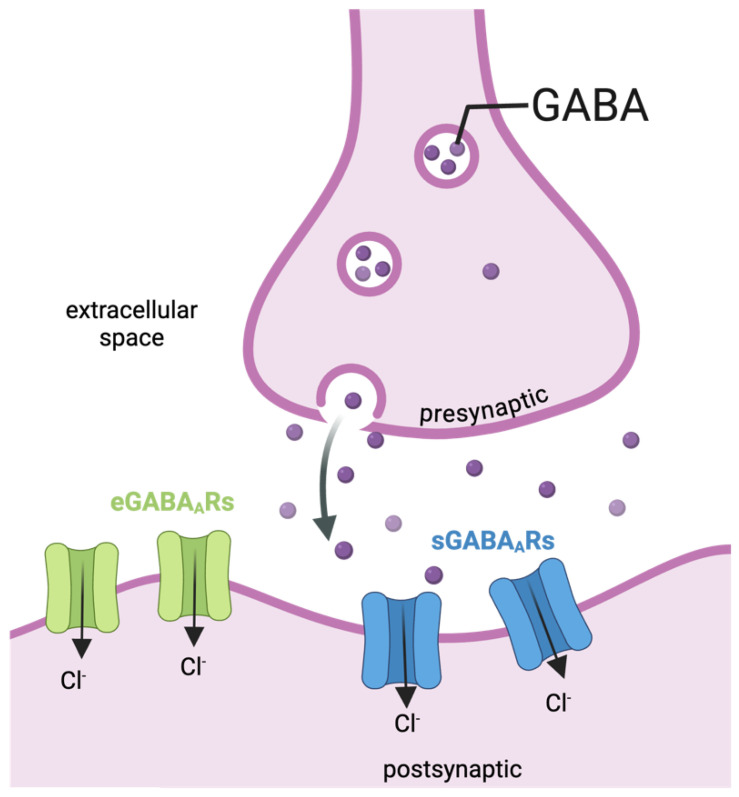
**Synaptic and extra-synaptic GABAergic transmission**. In the presynaptic terminal, the neurotransmitter GABA is packed within synaptic vesicles. When GABA is released into the synaptic cleft, it activates synaptic fast-phasic GABA_A_ receptors (sGABA_A_Rs, in blue). In the extra-synaptic zone, low ambient GABA concentration binds slow extra-synaptic GABA_A_ receptors (eGABA_A_Rs, in light green). Extra-synaptic GABA_A_ receptors desensitize less rapidly than synaptic GABA_A_ receptors, eliciting a constant tonic current in the neurons. Created with BioRender.com.

## Data Availability

Not applicable.

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
