# Peer review of "Unveiling GABA and Serotonin Interactions During Neurodevelopment to Re-Open Adult Critical Periods for Neuropsychiatric Disorders"

_ijms, 2025, doi:10.3390/ijms26125508_

Round 1
Reviewer 1 Report
Comments and Suggestions for Authors
Dear Authors,
Thank you for submitting your manuscript, "Unveiling GABA and Serotonin Interactions During Neurodevelopment to Reopen Adult Critical Periods for Neuropsychiatric Disorders". I am pleased to inform you that your work is potentially interesting and could be valuable to our readers. However, to ensure it meets the standards of our publication, I would like to offer some constructive feedback that I believe will help you improve the manuscript:
Introduction:
- Excessive background on GABA and serotonin can obscure new insights.
- Clearly outline research gaps addressed by the review.
- Define critical periods and provide timelines for brain regions.
Methodology:
- A summary table comparing key studies on GABA/serotonin and critical period modulation would enhance the review.
- While GABAergic mechanisms are well explained, the serotonergic system's role in critical period plasticity needs more discussion.
Discussion:
- The psychedelics section is overly lengthy; consider shortening it. A comparison table of different psychedelics (e.g., LSD, psilocybin, ketamine) on critical period plasticity would be beneficial.
- Avoid implying causation from correlation (e.g., increased spine density).
- Elaborate on how psychedelics interact with GABAergic and serotonergic circuits to reopen critical periods.
Figures:
- Add a schematic of key developmental stages, neurotransmitter actions, and treatment targets.
- To improve clarity, simplify the legends, and consider including a glossary for any acronyms used.
I hope you find this feedback helpful. I encourage you to consider these comments as you revise your manuscript. Thank you again for your submission, and I look forward to reviewing your revised version.
Best regards.
Needs revision.
Author Response
Please find the response in the attached file.

Reviewer 2 Report
Comments and Suggestions for Authors
The manuscript reviews the roles of GABA and serotonin in regulating brain plasticity of adult critical period in the developing and adult brain. Based on the recent findings, it shows that psychedelic substances can reopen critical period windows of plasticity in the adult brain, suggesting the potential of psychedelics drugs serving as promising therapeutic treatments for neuropsychiatric disorders in the nearly future. The summarization is appropriate potentially in terms of elucidating the core roles of neurotransmitters serotonin and GABA in neurodevelopment and brain plasticity. Although the manuscript is well written, due to the limited terms for selection, some minor points should be clarified by the authors before it can be recommended for publication.
1. In Figure 1, what is the difference between eGABAARs and sGABAARs ? Is it connected to the size of tonic current in the neurons?
2. In Figure 3a, there is an X next to the picture of SERT that should be clarified in the context of legend.
Author Response

(The authors gave the same response as above.)

Round 2
Reviewer 1 Report
Comments and Suggestions for Authors
Dear Authors,
I am fine with the current form.
Regards.